# Comparison of mortality and hospitalizations of older adults living in residential care facilities versus nursing homes or the community. A systematic review

Denis Boucaud-Maitre[1,2]*, Luc Letenneur[3], Moustapha Dramé[2,4], Nadine Taubé-Teguo[2,4], Jean-François Dartigues[3], Hélène Amieva[3], Maturin Tabué-Teguo[2,4]

1 Centre Hospitalier Le Vinatier, Bron, France, 2 Equipe EPICLIV, Université des Antilles, Fort-de-France, Martinique, 3 Inserm, U1219 Bordeaux Population Health Center, University of Bordeaux, Bordeaux, France, 4 Centre Hospitalo-Universitaire de Martinique, Fort-de-France, Martinique

* denis.boucaud@gmail.com

## Abstract

Residential care facility may provide a transition between living at home and a nursing home for dependent older people or an alternative to nursing homes. The objective of this review was to compare mortality and hospitalizations of older adults living in residential care facilities with those living in nursing homes or in the community. We searched Medline, Scopus and Web of Science from inception to December 2022. Fifteen cohort studies with 6 months to 10 years of follow-up were included. The unadjusted relative risk (RR) of mortality was superior in nursing homes than in residential care facilities in 6 of 7 studies (from 1.3 to 1.68). Conversely, the unadjusted relative risk of hospitalizations was higher in residential care facilities in 6 studies (from 1.3 to 3.37). Studies conducted on persons with dementia found mixed results, the only study adjusted for co-morbidities observing no difference on these two endpoints. Compared with home, unadjusted relative risks were higher in residential care facilities for mortality in 4 studies (from 1.34 à 10.1) and hospitalizations in 3 studies (from 1.12 to 1.62). Conversely, the only study that followed older adults initially living at home over a 10-year period found a reduced risk of heavy hospital use (RR = 0.68) for those who temporarily resided in a residential care facilities. There is insufficient evidence to determine whether residential care facilities might be an alternative to nursing homes for older people with similar clinical characteristics (co-morbidities and dementia). Nevertheless, given the high rate of hospitalizations observed in residential care facilities, the medical needs of residents should be better explored.

## Introduction

According to the World Health Organization (WHO), the number of people aged 60 and over will have overtaken the number of children under 5 by 2020 [1]. Population projections estimate that the proportion of people aged 60+ will almost double between 2015 and 2050, from 12% to 22% (or 2.1 billion people) [1]. The development of strategies for the care and housing

**Data Availability Statement:** All relevant data are within the paper and its Supporting information files.

**Funding:** The author(s) received no specific funding for this work.

**Competing interests:** The authors have declared that no competing interests exist.

of the older people, depending on their individual needs, is a priority. These individual needs depend on the physical and cognitive functions, their psychological state, their comorbidities and their social environment [2]. Older adults want to age at home and avoid institutionalization. The proportion of community-dwelling older people with functional limitations has increased in recent years [3]. Keeping these individuals at home often requires the implementation of medical (home care services, mobile geriatric teams, hospitalization at home) and other social aids (caregivers, meal delivery) which have an individual and collective cost. For older people who have severe medical and disability problems, the most widespread social model in developed countries remain nursing home care.

However, other housing models exist such as residential care facilities or foster families [4]. Residential care facilities (also called "senior housing", "independent living communities", "assisted living facilities", or "continual care retirement communities") have developed over the past decades. In general, each resident has a private apartment and access to common areas and services. Residential care facilities differ in size, type (residence or village), services offered and costs. These structures aim to promote the autonomy and social life of older people [5]. The socio-demographic and medical characteristics of residents in nursing homes and residential care facilities are not similar. Nevertheless, in the US, the age of residents is comparable and residential care facilities are increasingly admitting residents with functional limitations and/or Alzheimer's and other dementia. Indeed, the prevalence of Alzheimer's disease or other dementia in residential care facilities has increased from 5% in 2002 to 42% in 2010 [3].

Thus, determining the most appropriate care for dependent older adults is a public health priority. The place of residential care facilities, as an intermediate stage or final place of residence is therefore a key issue for the older adults, their families and health policies. Depending on the characteristics of the patient, existing models need to be properly assessed and compared to determine the effectiveness and cost-utility of each model. This is particularly true for older adults with dementia in the current debate over whether residential care facilities can substitute for, or delay the transfer to nursing homes. The effectiveness of residential care facilities, particularly in terms of mortality and hospitalizations, has been poorly studied in the literature. In 2012, a systematic review on patients with dementia found only one study suggesting that mortality and hospitalizations did not differ in residential care facilities and nursing homes [6]. The objective of this review was therefore to compare mortality and hospitalization rates reported among residents living in residential care facilities, nursing homes and/or in the community and to look for studies conducted on similar patients profiles with regard to dementia status and dependency.

## Methods

Our review was conducted in accordance with the Preferred Reporting Items for Systematic Reviews and Meta-analysis (PRISMA) guidelines [7]. The protocol for this review was registered in the International Prospective Register of Systematic Reviews (PROSPERO; CRD42022327207).

### Data source

We systematically searched Medline, Scopus, and Web of Science from inception up to 31 December 2022. We developed and conducted the literature search, using a combination of the MeSH terms "senior housing" or "independent living communities" or "residential care" or "continual care retirement" in the title or abstract and "mortality" or "death" or "hospitalizations" in the full text of the articles. Two reviewers (DBM, LL) screened the titles and abstracts. S1 Appendix provides the comprehensive search strategy used to identify original

research articles for inclusion in our systematic review. The same two reviewers independently assessed the full text of the articles for eligibility. Discrepancies were resolved by discussion. We also checked the reference lists of all reviews on this topic to identify articles that might have been missed. We limited the search to articles written in English and French.

## Eligibility criteria

**Population.** We only selected cohort studies conducted among older persons (≥65 years old).

**Intervention.** We selected studies that had a cohort design (prospective or retrospective cohort design) and at least six months of follow-up and that measured mortality or hospitalizations as outcomes. For duplicate publications from the same cohort, we selected those with the largest number of participants. We excluded cross-sectional studies and, reviews.

**Comparison.** We selected cohort studies comparing residential care facilities with nursing homes and/or communities.

**Outcomes.** To be included, we considered studies that reported the number of participants or person years and the number of deaths or hospitalizations in both groups (residential care facilities versus nursing home and/or communities).

## Data extraction and management

After the study selection process, one reviewer (DBM) extracted data from the original cohort studies. The characteristics extracted from each cohort were: name of the first author, year of publication, study design, length of follow-up, number of participants, mean age, percentage of dementia and percentage of participants with disability (functional status assessed by the Instrumental Activities of Daily Living (IADL) scale [8] or the Activities of Daily Living (ADL) scale [9]), number and percentage of deaths, number and percentage of hospitalizations and risk estimates.

## Quality assessment

The risk of bias was assessed using the Quality Assessment Tool for Observational Cohort and Cross-sectional studies [10], a recommended tool for analytic studies [11] by one reviewer (DBM). This process ensures that the quality of included studies is good enough to provide reliable results. Based on a series of questions, the goal was to identify potential flaws in the publication that could affect the measurement of the outcome. The quality of the studies was rated as "poor", "fair", or "good". Question 6 "For the analyses in this paper, were the exposure (s) of interest measured prior to the outcome(s) being measured?", question 7 "Was the time-frame sufficient so that one could reasonably expect to see an association between exposure and outcome if it existed?" and question 14 "Were key potential confounding variables measured and adjusted statistically for their impact on the relationship between exposure(s) and outcome(s)?" were considered critical because this review focuses on the relationship between mortality/hospitalization and types of accommodations. Potentially confounding variables affecting the results were age, gender, comorbidities, levels of dependency and cognitive function. In cases where studies answered "no" to questions 7 and 14, quality was rated as "poor".

## Statistical analysis

Unavailable relative risks (RR) and confidence intervals were calculated from the number of events and the number of residents for each study. After reviewing each of the studies included in this review, we considered that a meta-analysis was not relevant, due to the low

methodological quality of the studies, the diversity of study designs and follow-up and the heterogeneity of populations, for which essential baseline characteristics were not available.

## Results

Our search yielded 8,716 records, of which 6,975 remained after eliminating duplicates (Fig 1). When screening tittles and abstracts, 6,918 records were excluded, leaving 24 full texts to be assessed. Nine studies were excluded, among them two focused on specific mortality endpoints (respiratory mortality and suicide [12, 13]), one related to emergency department visits rather than hospital admission [14], one pooled data from residential care facilities and nursing home data [15], three were systematic review [6, 16, 17] and two were sub-studies of other studies. A total of 15 studies met the eligibility criteria and were included in this review [18–32].

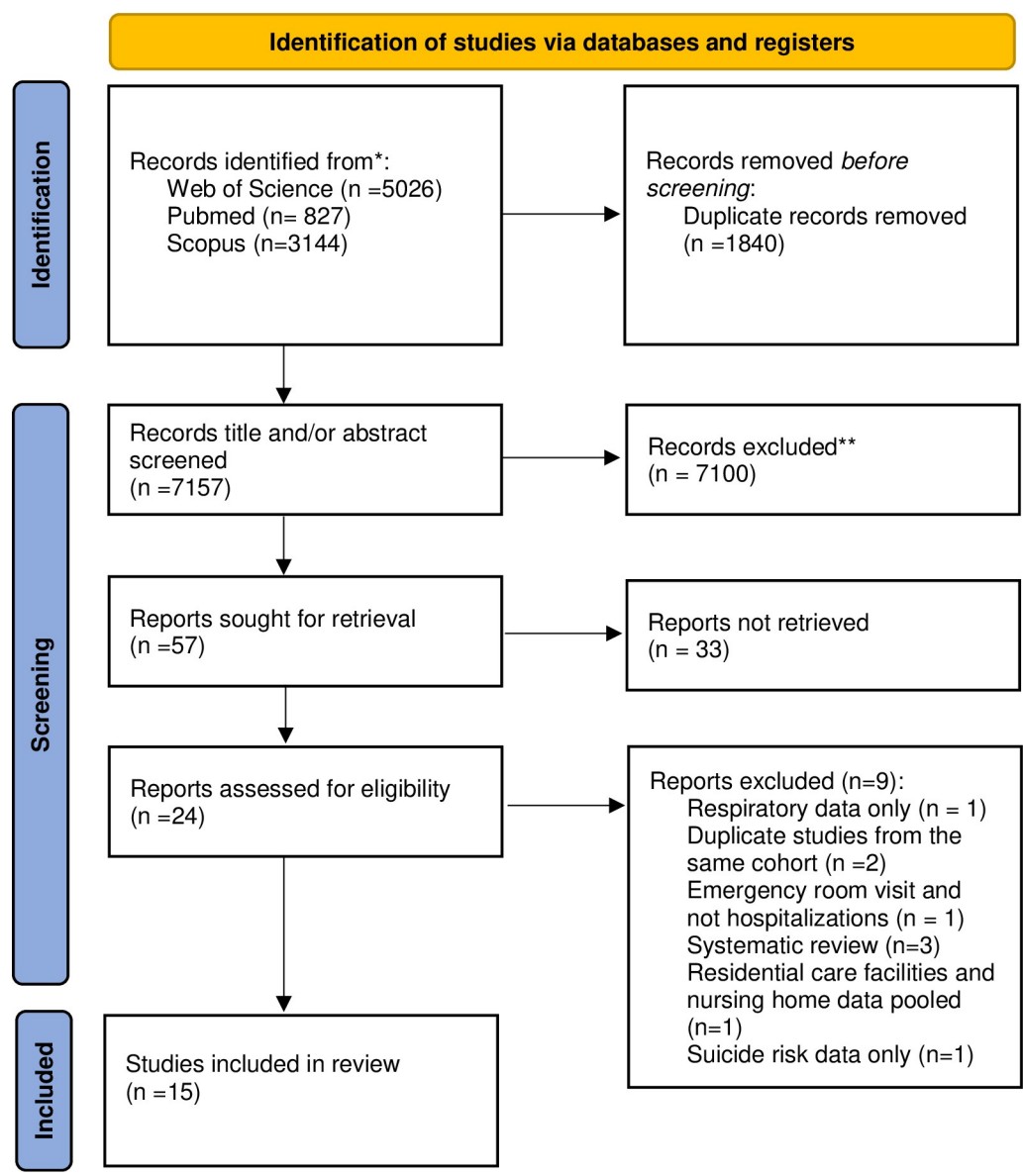

**Fig 1. Flow diagram of included studies.**

## Characteristics of the included studies

The studies were published between 2000 and 2020, 5 studies were prospective cohort studies [18, 19, 21, 23, 24] and 10 were retrospective cohort studies [20, 22, 25–32]. The number of participants ranged from 158 to 691,388 for the studies comparing residential care facilities with nursing homes, and from 808 to 3,366,303 for the studies comparing residential care facilities to home-based care. The studies follow-up ranged from 6 months to 10 years. In three studies [22, 25–26], age was not available, and in the remaining 12 studies, the mean age of participants was greater than 80 years or a majority of participants were 75 and older. For eight studies, MMSE score or the percentage of dementia among participants were not available for one or the both groups of participants. Three studies comparing residential care facilities to nursing homes focused only on dementia patients [19–21]. For the other 5 studies, the oldest adults with dementia were reported in nursing homes rather than in residential care facilities in two studies (72.7% versus 30.3% [20], MMSE score = 17.88 versus 16.03 [24]), there was no difference for one study (MMSE score = 23.3 versus 23.1 [18] and one study reported oldest adults with dementia in residential care facilities rather than at home (33.9% versus 8.4%) [31]. Functional ability was available for three studies, all comparing residential care facilities to nursing homes, with a lower disability score in nursing homes than in residential care facilities [18, 19, 24]. Six studies assessed cohorts from US, 3 from Canada, 3 from United Kingdom, 1 from Ireland, Taiwan and Australia.

## Risk of bias

Of the 15 studies, two were rated "good" [18, 31], six "fair" [18, 20, 22, 28, 29, 32] and seven "poor" [21, 24–27, 30] (S1 Appendix). Quality was considered good when similar clinical baseline characteristics were observed or when adjustments on age, sex, dementia and dependency (question 14) were implemented. We considered that failure to adjust for any of this variables downgraded the quality of the study to "fair", no adjustment or insufficient time (inferior to one year) were rated as "poor" studies (S1 Annex).

**1. Comparison of mortality between residential care facilities and nursing homes.** Nine studies compared mortality between residential care facilities and nursing homes [18–26] (Table 1). Six of the seven studies with at least one year of follow-up described a higher unadjusted relative risk in nursing homes than in residential care facilities, ranging from 1.3 to 1.68. The only other study with more than 1 year of follow-up suggesting a lower risk [26] had in-hospital mortality, not total mortality as its endpoint.

Three studies [19–21] focused specifically on patients with dementia and a fourth adjusted for dementia [18]. The study by Thomas *et al.* [20], comparing 88,867 residents in residential care facilities with 602,521 residents of nursing homes (1-year follow-up) reported a risk of 1.58 [1.56–1.6]. The study by Sloane *et al.* [19] found no significant difference in mortality between residential care facilities and nursing homes for mild dementia or moderate/severe dementia after adjustment for age, gender, race, education, marital status, length of stay, cognition and comorbidities. Finally, the study by Shah *et al.* [22] found a gender and age-adjusted ratio of 419 (396–442) for nursing homes alone versus 284 (266–302) for residential care facilities. Further standardization for dementia diagnosis reduced the ratio to 309 (292–326) and to 218 (205–232), respectively.

**2. Comparison of hospitalizations between residential care facilities and nursing homes.** Seven studies [19–21, 26–29] compared hospitalization rates between residential care facilities and nursing homes (Table 2). All studies were North American (3 in Canada, 3 in the US) with an exception of one study from the UK. All described an unadjusted increased relative risk of hospitalizations in residential care facilities compared with nursing homes, ranging

**Table 1.  Studies comparing residential care facilities versus nursing homes on mortality.**

| Author, year, Country | Study design and follow-up | Participants/cases | Age (mean or %), years | Percentage of dementia | Percentage of mortality | Unadjusted relative risk | Adjusted risk ratio or hazard ratio |
|---|---|---|---|---|---|---|---|
| Pruchno, 2000, US [18] | Prospective, 15 months | • Nursing Homes: 76 • Residential care facilities:82 | • Nursing Homes: 87.4 • Residential care facilities: 86.2 | • Nursing Homes: MMSE score = 23,3 • Residential care facilities: MMSE score = 23.1 | • Nursing Homes: 18.4% (14/76) • Residential care facilities: 12.2% (10/82) | 1.51 [0.71–3.19] | No |
| Sloane, 2005, US [19] | Prospective, 1 year | • Nursing Homes: 479 • Residential care facilities: 773 | • Nursing Homes:: 84.9 • Residential care facilities: 84.4 | • Nursing Homes: Moderate to severe: 49.3% • Residential care facilities: Moderate to severe: 29.4% | • Nursing Homes: 22.5% (108/479) • Residential care facilities: 14.7% (114/773) | 1.53 [1.21–1.94] | Incidence rate per 100 participants (mild dementia): nursing homes: 4.2 versus residential care facilities: 3.2 (not-significant) Moderate to severe dementia: 4.2 versus 3.7 (non-significant) Incidence rate adjusted for baseline age, gender, race, education, marital status, length of stay, cognition and number of comorbidities |
| Thomas, 2020, US [20] | Retrospective, 1 year | • Nursing Homes: 602,521 • Residential care facilities: 88,867 | • Nursing Homes: >75 years: 80.6% • Residential care facilities: >75 years: 88.2% | • Nursing Homes: 100% • Residential care facilities: 100% | • Nursing Homes: 31.1% (187350/602521) • Residential care facilities: 19.7% (17491/88867) | 1.58 [1.56–1.6] | No |
| Resnick, 2015, US [21] | Prospective, 6 months | • Nursing Homes: 103 • Residential care facilities: 93 | • Nursing Homes: 83.7 • Residential care facilities: 85.7 | • Nursing Homes: 100% (MMSE score: 8.7) • Residential care facilities: 100% (MMSE score: 5.8) | • Nursing Homes: 0% (0/103) • Residential care facilities: 0% (0/93) | - | No |
| Shah, 2013, England and Wales [22] | Retrospective, 1 year | • Nursing Homes: 4109 • Residential care facilities: 4320 | Not specified | Nursing Homes and residential care facilities combined: 38,9% | Nursing Homes: 30.8% (1265/4109) Residential care facilities: 22.3% (963/4320) | 1.38 [1.28–1.48] | Age and sex-adjusted hazard ratios: 1.48. The ratio for nursing homes alone was 419 (396–442) and that for residential homes was 284 (266–302). Further standardization for dementia diagnosis reduced the ratio to 309 (292–326) for nursing homes and to 218 (205–232) for residential homes. |
| McCann, 2009, Ireland [23] | Prospective, 5 years | • Nursing Homes: 895 • Residential care facilities: 577 | >75 years: • Nursing Homes: 88% • Residential care facilities: 89% | Not specified | • Nursing Homes: 70% (626/895) • Residential care facilities: 54% (311/577) | 1.3 [1.19–1.42] | No |
| Liu, 2010, Taiwan [24] | Prospective, 9 months | • Nursing Homes: 140 • Residential care facilities: 185 | >75 years: • Nursing Homes: 65% • Residential care facilities: 73% | • Nursing Homes: MMSE score = 17.88±8.91 • Residential care facilities: MMSE score = 16.03±6.90 | • Nursing Homes: 0% (0/140) • Residential care facilities: 3.2% (6/185) | - | No |

(*Continued*)

**Table 1.** (Continued)

| Author, year, Country | Study design and follow-up | Participants/ cases | Age (mean or %), years | Percentage of dementia | Percentage of mortality | Unadjusted relative risk | Adjusted risk ratio or hazard ratio |
|---|---|---|---|---|---|---|---|
| Rothera, 2002, UK [25] | Retrospective, 20 months | • Nursing Homes: 499 • Residential care facilities: 866 | Not specified | Not specified | • Nursing Homes: 39.1% (195/499) • Residential care facilities: 23.3% (202/866) | 1.68 [1.42–1.97] | No |
| Godden, 2001, UK [26] | Retrospective, 1 year | • Nursing Homes: 1700 • Residential care facilities:1504 | Not specified | Not specified | • Nursing Homes: 5.7% (97/1700) • Residential care facilities: 8.8% (133/1504) | 0.65 [0.5–0.83] (Hospital death and not total mortality) | No. |

from 1.3 to 3.37, except for one small study. Five studies had 1 year follow-up, with hospitalization rates of 30% to 40% in residential care facilities versus 10% to 30% in nursing homes.

Four studies included only residents with dementia. The study with the highest relative risk was conducted in Canada [28]. It found hospitalization rates of 36.1% in residential care facilities versus 10.7% in nursing homes. Less severe cognitive impairment (Hazard Ratio: 0.35 [0.18–0.67]) was associated with a lower hospitalization rate in this study. The study by Thomas *et al*. [20] reported a relative risk of 1.30 [1.29–1.31]. Conversely, in the study by Sloane *et al*. [19] adjusted for age, gender, ethnicity, education, marital status, length of stay, cognition and number of comorbidities, the risk of hospitalization was higher for patients with mild dementia (14.2% versus 8.4%, p = 0.009) in residential care facilities but not for those with moderate or severe dementia (14.2% versus 10.0%, p = 0.115).

**3. Comparison of mortality between residential care facilities and the community.** Four studies have been conducted to compare mortality of older patients living in residential care facilities and the community (Australia [30], Ireland [23], two in the US [20, 31]) (Table 3). All suggest a higher mortality rate in residential care facilities, with unadjusted RRs ranging from 1.34 to 10.1. In these four studies, the characteristics of the elderly differed by age and/or dementia between the two groups. The Australian study [30] comparing 3,330,987 older people at home versus 35,316 residents in residential care facilities observed 1-year mortality rates of 3.6% versus 34.6% respectively, giving an age and gender adjusted odd-ratio (OR) of 10.1 (95% CI: 9.8–10.5). The Irish study [23] found an OR of 1.63, adjusted for age, sex, general health and marital status with a 5-year follow-up. Bartley's study [30] retrieved an OR of 2.4, adjusted for Charlton Comorbidity Index and marital status. Finally, Thomas *et al.'s* study [20] of patients with dementia reported an unadjusted relative risk of 1.34 [1.33–1.36].

**4. Comparison of hospitalizations of older adults living in residential care facilities and the community.** Three 1-year longitudinal studies from the US or the UK suggest a greater risk of hospitalization of older adults living in residential care facilities than in the community ranging from 1.12 to 1.65 [20, 26, 31] (Table 4). One-year hospitalization rates ranged from 31 to 48%. Only one study, by Park *et al*. [32], followed elderly people initially living at home and compared the 10-year hospitalization rate between those who went to residential care facilities and those who did not. In this study, the risk of hospitalization among those entering residential care facilities was decreased for heavy hospital use (RR 0.68 (p<0.001)) but not for moderate hospital use.

**Table 2. Studies comparing hospitalizations between residential care facilities and nursing homes.**

| Author, year, Country | Study design and follow-up duration | Participants/ cases | Age (mean and %), years | Percentage of dementia | Percentage of hospitalizations | Unadjusted relative risk | Adjusted risk ratio or hazard ratio |
|---|---|---|---|---|---|---|---|
| McGregor, 2014, Canada [27] | Retrospective, 3 years | • Nursing homes:12209 • Residential care facilities:842 | • Nursing homes: 83.1 • Residential care facilities: 81.5 | Not specified | • Nursing homes: 42.0% (5125/12209) • Residential care facilities: 69.1% (582/842) | 1.65 [1.57–1.73] | No |
| Maxwell, 2015, Canada [28] | Retrospective, 1 year | • Nursing homes: 691 • Residential care facilities: 609 | • Nursing homes: 86.4 • Residential care facilities: 85.7 | • Nursing homes: 100% • Residential care facilities:: 100% | • Nursing homes: 10.7% (74/691) • Residential care facilities: 36.1% (220/609) | 3.37 [2.65–4.29] | No |
| Hogan, 2014, Canada [29] | Retrospective, 1 year | • Nursing homes: 976 • Residential care facilities: 1066 | • Nursing homes: Not specified • Residential care facilities: 84.9 | • Nursing homes: not specified • Residential care facilities:: 57.1% | • Nursing homes: 14.0% (137 /976) • Residential care facilities: 38.7% (413/1066) | 2.76 [2.32–3.28] | No |
| Sloane, 2005, US [19] | Prospective, 1 year | • Nursing homes: 479 • Residential care facilities: 773 | • Nursing homes: 84.9 • Residential care facilities: 84.4 | • Nursing homes: Moderate to severe: 49.3% • Residential care facilities:: Moderate to severe: 29.4% | Mild dementia: • Nursing homes: 8.4% (20/243) • Residential care facilities: 14.2% (78/546) Moderate to Severe dementia: • Nursing homes: 10.0% (24/236) • Residential care facilities: 14.2% (32/227) | 1.55 [1.11–2.16] | Mild dementia: P = 0.009 adjusted for baseline age, gender, race, education, marital status, length of stay, cognition and number of comorbidities. Moderate to severe dementia: P = 0.115 adjusted for baseline age, gender, race, education, marital status, length of stay, cognition and number of comorbid conditions. |
| Thomas, 2020, US [20] | Retrospective, 1 year | • Nursing homes: 602521 • Residential care facilities: 88867 | • Nursing homes: >75 years: 80.6% • Residential care facilities: >75 years: 88.2% | • Nursing homes: 100% • Residential care facilities: 100% | • Nursing homes: 29% (174323/602521) • Residential care facilities: 37.6% (33457/88867) | 1.30 [1.29–1.31] | No |
| Godden, 2001, UK [26] | Retrospective, 1 year | • Nursing homes: 1132 • Residential care facilities:1504 | Not specified | Not specified | • Nursing homes: 22,3% (253/1132) • Residential care facilities: 31,2% (469/1504) | 1.4 [1.22–1.59] | No |
| Resnick, 2015, US [21] | Prospective, 6 months | • Nursing homes: 103 • Residential care facilities: 93 | • Nursing homes: 83.7 • Residential care facilities: 85.7 | • Nursing homes: 100% (MMSE score: 8.7) • Residential care facilities: 100% (MMSE score: 5.8) | • Nursing homes: 0% (0/103) • Residential care facilities: 0% (0/93) | - | No |

## Discussion

In this literature review, we analyzed 15 studies comparing older adults living in residential care facilities with older adults living in nursing homes or the community.

In general, a twofold increase in mortality was observed in nursing homes compared to residential care facilities. This result was expected since nursing homes generally accommodate patients at the end of life, with significant co-morbidities and a severe degree of dependency. We observed that the age of patient was generally similar in studies comparing nursing homes

**Table 3. Studies comparing mortality of older adults living in residential care facilities and the community.**

| Author, year, Country | Study design | Participants/ cases | Age (mean and %), years | Percentage of dementia | Percentage of mortality | Unadjusted relative risk | Adjusted risk ratio or hazard ratio |
|---|---|---|---|---|---|---|---|
| Inacio, 2020, Australia [30] | Retrospective, 1 year | • Community: 3330987 • Residential care facilities: 35316 | >75 years: • Community: 63.0% • Residential care facilities: 88.2% (mean: 85 years) | Not specified | • Community: 3.6% (119815/3330987) • Residential care facilities: 34.6% (12225/35316) | 9.62 [9.48–9.77] | OR: 10.1 (95% CI: 9.8–10.5) adjusted by sex and age |
| Bartley, 2018, USA [31] | Retrospective, with age- and sex-matched, 1 year | • Community: 404 • Residential care facilities: 404 | • Community: 86.8 • Residential care facilities: 86.8 | Community: 8.4% Residential care facilities: 33.9% | • Community: 9.4% (38/404) • Residential care facilities: 20.3% (82/404) | 2.16 [1.51–3.09] | OR: 2.4 Adjusted for Charlson Comorbidity Index and marital status |
| McCann, 2009, Ireland [23] | Prospective, 5 years | • Community: 205566 • Residential care facilities: 577 | >75 years: • Community: 42% • Residential care facilities: 89% | Not specified | • Community: 22% (45224/205566) • Residential care facilities: 54% (311/577) | 2.45 [2.27–2.64] | HR: 1.63 (1.44–1.85) Adjusted for age, sex, general health and marital status |
| Thomas, 2020, US [20] | Retrospective, 1 year | • Community: 2074420 • Residential care facilities: 88867 | >75 years: Community: 80.6% Residential care facilities: >75 years: 88.2% | • Community: 6.1% • Residential care facilities: 30.3% | • Community: 14.7% (303891/2074420) • Residential care facilities: 19.7% (17491/88867) | 1.34 [1.33–1.36] | No |

and residential care facilities, suggesting that co-morbidities may have a greater impact on mortality than biological age [33]. Unfortunately, in several studies, comorbidities, in particular the level of dependency or severity of dementia, were poorly documented and/or not considered, except in the study by Sloane *et al*. The studies included in the review also reported an

**Table 4. Studies comparing hospitalizations of older adults living in residential care facilities and the community.**

| Author, year, Country | Study design | Participants/ cases | Age (mean and %), years | Percentage of dementia | Percentage of hospitalizations | Unadjusted risk ratio | Adjusted risk ratio or hazard ratio |
|---|---|---|---|---|---|---|---|
| Park, 2018, US [32] | Retrospective, 10 years | • Community: 975 • Residential care facilities: 214 | • Community: 82.4 • Residential care facilities: 83.3 | Not specified | Not specified | | RR: 0.68 (p<0.001) for heavy hospital use RR: 0.89 (NS) for moderate hospital use. Adjusted with death, sociodemographics, health, social support, regions |
| Bartley, 2018, USA [31] | Retrospective, with age- and gender-matched, 1 year | • Community: 404 • Residential care facilities: 404 | • Community: 86.8 • Residential care facilities: 86.8 | • Communities: 8.4% • Residential care facilities: 33.9% | • Community: 31.4% (127/404) • Residential care facilities: 48.3% (195/404) | 1.54 [1.29–1.83] | OR: 2.03 [CI: 1.5–2.7] Adjusted for Charlson Comorbidity Index and marital status |
| Thomas, 2020, US [20] | Retrospective, 1 year | • Community: 2074420 • Residential care facilities: 88867 | >75 years: 80.6% Residential care facilities: 88.2% | • Communities: 6.1% • Residential care facilities: 30.3% | • Community: 33.6% (697968/2074420) • Residential care facilities: 37.6% (17491/88867) | 1.12 [1.11–1.13] | No |
| Godden, 2001, UK [26] | Retrospective, 1 year | • Community: 83606 • Residential care facilities:1504 | Not specified | Not specified | • Community: 18.9% (15239/80402) • Residential care facilities: 31.2% (469/1504) | 1.65 [1.52–1.78] | No |

increased risk of hospitalizations in residential care facilities compared to nursing home, with hospitalization rates of about 30% per year in residential care facilities. From a public health perspective, the cost of these hospitalizations is important to consider when evaluating the efficiency of this model. The causes of hospitalizations could be different between these two settings, especially concerning falls [34, 35] or polymedication. Targeted geriatric interventions such as telemonitoring [36] could reduce avoidable hospitalizations in residential care facilities. Moreover, this high risk of hospitalization raises questions about the ability of residential care facilities to meet the medical needs of older adults. In the US, only 48.2% of community-based residential settings offer a range of services including nursing care, medication assistance, meals, laundry, cleaning, transportation, and recreation and 29.1% have access to all these services except for nursing care and medication assistance [37]. It seems important to better define the clinical profile of the older adults who may be candidates for residential care facilities. Indeed, residential care facilities are often considered an appropriate setting for cognitively impaired patients in the US and Canada. In these countries, a high rate of residents suffers from dementia (58% in the Canadian study by Maxwell *et al.* [28] or 68% of individuals in the American study by Watson *et al.* [38]). Yet dementia is reported to be the most common predisposing factor (>90%) that precipitates the move of older adults to an assisted living or nursing home [39]. Previous research indicates that the percentage of facilities that provide staff training related to psychiatric disorders in older adults is low and generally inadequate in the US [40]. Increased medical and nursing support may be an option to reduce hospitalization rates.

The vast majority of studies have been conducted in the US and Canada. In other countries, the clinical characteristics of residents may be different in residential care facilities. From this point of view, residential care facilities could be a step when patients at home require more care or become frail, before the development of severe cognitive impairment [40]. In the UK, the proportion of resident with severe dementia in residential care facilities appears to be low (2.1% versus 24.1% in nursing homes [41]). In Sweden, only 20% of residents suffered from dementia [42]. In France, residents in residential care facilities tend to be frail (53.7%), without being systematically disabled (mean ADL score; SD = 5.4; 0.9) and the role of residential care facilities is rather to address social isolation, social vulnerability, and loneliness [43]. Further studies in Europe are needed to determine whether the hospitalization rate is comparable to that of the North American studies.

Compared to living at home, mortality in residential care facilities was much higher, with RR ranging from 1.34 to 10.1. Again, the methodological quality of these studies was critical, as none of them at least adjusted for dementia status or level of dependency. Studies are needed to compare home care systems with residential care facilities for older adults suffering from social isolation or comorbidity. Indeed, several devices have been developed over time to promote home care, such as remote monitoring, telemedicine, or home care services [44, 45]. In particular, telemedicine has shown encouraging results in the management of care, prevention or management of chronic pathologies (particularly cardiovascular or diabetes) or adherence to medication [46]. With regards to hospitalizations, a higher risk was found in residents living in residential care facilities compared to those living at home in 3 of the 4 studies with a follow-up of one year. However, the study by Park *et al.* [32] contrasts with these results, both because of the methodological quality of the study and because of the results obtained. This study suggests that residential care facilities may reduce the risk of major hospitalization, based on a cohort of patients initially at home, some of whom may or may not enter a residential care facilities during the 10 years of follow-up of the study.

The main limitation of this review is the methodological quality of the studies. No study compared these models of care up to institutional entry with minimal adjustment for age,

gender, dementia, and dependency, and only the study by Park *et al.* [32] analyzed the health trajectory over time of older adults initially living at home. The medical characteristics of residents in terms of dementia and standardized cognitive assessment, activities of daily living, frailty and length of stay were poorly described. Moreover, the type of residential care facilities and services offered may vary from country to country or from institution to institution [47, 48]. Finally, socioeconomic characteristics (marital status, income, etc.), which may influence the choice of institution (public or private), have rarely taken into account in the studies. For all these reasons, we considered that meta-analyses were not relevant because the studies did not compare patients with similar characteristics profile. The level of evidence for the effectiveness of residential care facilities on mortality and hospitalizations compared to nursing home or communities is therefore low. Nevertheless, the particularly high rate of hospitalization in residential care facilities raises the question of the lack of medical and paramedical staff and the cost of these hospitalizations. Nevertheless, the potential benefits of residential care facilities versus nursing homes or home care are not limited to mortality and hospitalizations. The effects on physical function, quality of life, happiness, cognition and other aspects of health would need to be compared. A final limitation is that the search strategy was limited to Medline, Scopus and Web of Science and to articles written in English or French. Nevertheless, It has been established that the exclusion of non-English language articles has only a minimal effect on the overall conclusions of the reviews [49]. A single reviewer conducted the data extraction and quality assessment of the studies, which may reduce the diversity of the studies included.

## Conclusion

This systematic review raises important clinical and policy questions. The place of residential care facilities in the health care pathway of older adults, as an intermediate or alternative step to home or nursing home, has not been sufficiently studied. Although patient's profiles are likely to differ and care systems are not identical across the world, the particularly high rate of hospitalizations in these settings requires further investigations to assess the effectiveness and efficiency of this model. If residential care facilities are considered as an alternative for older people with mild to moderate dementia, studies of good methodological quality have to be implemented. Preventive and palliative care, depending of levels and types of medical, functional, and psychosocial needs, may be useful to reduce avoidable hospitalizations.

## Supporting information

**S1 Checklist. PRISMA 2020 checklist.**
(DOCX)

**S1 Appendix. Search strategy.**
(DOCX)

**S1 Annex. Quality studies.**
(DOCX)

## Author Contributions

**Conceptualization:** Denis Boucaud-Maitre, Jean-François Dartigues, Hélène Amieva, Maturin Tabué-Teguo.

**Formal analysis:** Denis Boucaud-Maitre.

**Investigation:** Denis Boucaud-Maitre, Luc Letenneur.

**Methodology:** Denis Boucaud-Maitre, Jean-François Dartigues, Hélène Amieva.

**Supervision:** Jean-François Dartigues, Maturin Tabué-Teguo.

**Validation:** Luc Letenneur, Moustapha Dramé, Jean-François Dartigues, Hélène Amieva, Maturin Tabué-Teguo.

**Writing – original draft:** Denis Boucaud-Maitre.

**Writing – review & editing:** Luc Letenneur, Moustapha Dramé, Nadine Taubé-Teguo, Jean-François Dartigues, Hélène Amieva, Maturin Tabué-Teguo.

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
