## [Decision Letter · Decision Letter 0]

28 Mar 2023

PONE-D-23-05379Comparison of mortality and hospitalizations of older people living in residential care facilities versus nursing homes or communities. A systematic review.PLOS ONE

Dear Dr. Boucaud-Maitre,

Thank you for submitting your manuscript to PLOS ONE. After careful consideration, we feel that it has merit but does not fully meet PLOS ONE’s publication criteria as it currently stands. Therefore, we invite you to submit a revised version of the manuscript that addresses the points raised during the review process. Please submit your revised manuscript by May 12 2023 11:59PM. If you will need more time than this to complete your revisions, please reply to this message or contact the journal office at plosone@plos.org. Please include the following items when submitting your revised manuscript:A rebuttal letter that responds to each point raised by the academic editor and reviewer(s). You should upload this letter as a separate file labeled 'Response to Reviewers'.A marked-up copy of your manuscript that highlights changes made to the original version. You should upload this as a separate file labeled 'Revised Manuscript with Track Changes'.An unmarked version of your revised paper without tracked changes. You should upload this as a separate file labeled 'Manuscript'.

We look forward to receiving your revised manuscript.

Kind regards,

Mickael Essouma, M. D.

Academic Editor

PLOS ONE

Journal Requirements:

2. We note you have included a table to which you do not refer in the text of your manuscript. Please ensure that you refer to Table 3 and 4 in your text; if accepted, production will need this reference to link the reader to the Table.

Reviewers' comments:

Reviewer's Responses to Questions

**Comments to the Author**

1. Is the manuscript technically sound, and do the data support the conclusions?

Reviewer #1: Yes

Reviewer #2: No

2. Has the statistical analysis been performed appropriately and rigorously? 

Reviewer #1: N/A

Reviewer #2: No

3. Have the authors made all data underlying the findings in their manuscript fully available?

Reviewer #1: Yes

Reviewer #2: No

4. Is the manuscript presented in an intelligible fashion and written in standard English?

Reviewer #1: Yes

Reviewer #2: No

5. Review Comments to the Author

Reviewer #1: The manuscript is based on a systematic literature review of an important topic, namely studies comparing either mortality or hospitalizations of older people living in residential care facilities versus nursing homes or communities. According to my evaluation the analysis has been carried out correctly and the manuscript is well written so I can recommend publication. The manuscript leaves the impression of only a limited number of available high quality studies and hence, the conclusion is wisely cautiously formulated. The Discussion is thorough and all central limitations are mentioned. The only detail that caught my attention was that study 19 is classified as fair but nevertheless, is described as lacking data on age. In this research area I find that the age distribution is of central importance. Second, I mention probably a typo as abbreviations are consistently used but on page 10 for some reason 'nursing home' is written out, although the abbreviation NH has already earlier been introduced.

Reviewer #2: Peer review for the article: "Comparison of mortality and hospitalizations of older people living in residential care facilities versus nursing homes or communities. A systematic review."

1. Recommendation

Manuscript ref no. PONE-D-23-05379

Major revision.

2. Comments

2.1. General comment

The authors have attempted a systematic review of the literature (SLR) to assess the impact (the effectiveness) of the care delivered to elderlies at residential care facilities (compared to that delivered at their homes and in nursing homes [exposures]0 on elderlies future health outcomes (all-cause hospitalizations and case fatality). However, this message is not clearly stated throughout the manuscript, and the methods used are highly questionable. There is a need for a major revision of the SLR and the manuscript.

2.2. Specific comments

2.2.1. Major comments

2.2.1.1. Introduction

This is a global SLR, but not s SLR of COVID-19 or US-based data. So, there is no reason why you emphaisze on COVID-19 and USA in the introduction. I understand that the demand could have increased with the advent of the COVID-19 pandemic (West et al. Age and Ageing, Volume 50, Issue 2, March 2021, Pages 294–306, https://doi.org/10.1093/ageing/afaa289), but this could just be mentioned within the text as one among the many reasons why it is more and more important now to address the importance of residential care facilities when deciding about the place of care for elderlies.

Your introduction should not exceed one page, although it is important that you revise the current introduction providing information to these important questions (obviously for the global population), because PLOS One is a generalist not specialist journal: Is the health of older people a concern in the society? Notably, what is the proportion of older people in the current population and how will change in the coming decades? In general, how is the health state of older people: disease, disability-adjusted life years, years lived with disease, most frequent diseases, comorbidities, drug-consumption level? What are the available care delivery systems for older people? What is the place of residential care facilities in those delivery systems? How have the covid-19 pandemic and other stressors increased the demand for care delivery systems (including residential care facilities) in recent years? What should be done for a remedial to this alarming situation? How will your SLR help to remedy to the situation: is it intended to inform older people health experts? Stakeholders? End with a clear statement of the aim of your SLR. Writing and subsequent editing, will help you to deliver all those important messages within a short text of one page.

2.2.1.2. Methods

The reference 6 is irrelevant, because the registration number from PROSPERO is sufficiently informative.

Provide a reference for the PRISMA guidelines used. the latest one which is better to provide is: Page et al. BMJ 2021; 372 doi: https://doi.org/10.1136/bmj.n71.

The searched databases: I advise to include those that capture data for usually forgotten regions of Africa (e.g. AJOL, Africa Index Medicus), Asia (see DOI: 10.1097/EDE.0000000000000325) and South America (e.g. LILACS), as a global SLR should fairly provide data from across the globe. I would also include EMBASE (see Bramer et al. J Med Libr Assoc. 2018 Oct; 106(4): 531–541. doi: 10.5195/jmla.2018.283).

The search terms should be revised. The search strategy should be based on three main concepts: the population of interest (elderly people: all the terms referring to this phrase and available in the searched databases should be sought), the exposure (residential care facilities again with all the terms referring to it, the nursing home with all synonyms, the home-based care with all synonyms) and the outcomes (hospitalizations with all synonyms such as admission, emergency visit; case fatality with all synonyms such as mortality and death). Did you perform hand searching? This should be clearly stated and explained in the manuscript. Work with an expert librarian.

Inclusion and exclusion criteria: Do not place limits on the language of articles because this increases the selection bias, and the study is funded. So, you can request help from certified translators for articles written in languages which you are not familiar with, or use websites such as deepl translator. I am concerned by the fact that you excluded articles that focused on specific diseases, but you focused on dementia. Can you explain this inconsistency ? (which should be resolved throughout the manuscript). I am also concerned by the fact that you compare the outcome resulting from the care at residential care facility with that from original homes and nursing homes, but you exclude cross-sectional analytical studies that can also do this well: is there a justification? As stated above, it is also important that you clarify for authors the exposure and the outcomes assessed. Along this line, I would use the term "case fatality" throughout the text, instead of "mortality" (see Kelly and Cowling. Epidemiology 24(4):p 622-623, July 2013. | DOI: 10.1097/EDE.0b013e318296c2b6).

What do the IADL and ADL scores mean (page 4, line 86)?

-The data extraction should be revised as well, and the standardized data abstraction sheet uploaded as supplemental material with your submission. These are mandatory data for high-quality epidemiological studies: type of sampling, setting [community versus hospital-based, registry], response rate [for surveys], locality [urban versus rural vs semi-urban], region of origin [based on which classification: world bank? UNSD?], country of origin, study design (please, go through this paper to gain an insight on types of traditional epidemiological observational studies . It should be noted that cohorts are always longitudinal studies, so the repetition in the text is not warranted), timing of data collection (retrospective/prospective/ambispective i.e., prospective + retrospective. Note that the mode of data collection is more important than when authors collected data i.e., in a registry with prospectively collected data but which can be retrospectively consulted by authors of a given manuscript, it is the fact that data were collected prospectively which is relevant; and this type of study is different from a retrospective chart review [Vassar and Holzmann. J Educ Eval Health Prof 2013, 10: 12 • http://dx.doi.org/10.3352/jeehp.2013.10.12.] where data were collected as usual in the clinic without aim to conduct a study, but subsequently some researchers conduct a study with those data that they therefore collect retrospectively).

References 7 and 8 should be omitted because they are irrelevant and outdated. The tool used for quality assessment (choose anyone reliable that you are comfortable with among those provided here: Munn et al. Int J Health Policy Manag

. 2014 Aug 13;3(3):123-8. doi: 10.15171/ijhpm.2014.71.) should be clearly stated, and how you did the assessment should be illustrated in tables for each study, in the supplemental material.

There is a need for a meta-analysis. The between-study heterogeneity is a conditional limitation of the meta-analysis (see Valentine et al. Journal of Educational and Behavioral Statistics April 2010, Vol. 35, No. 2, pp. 215–247 DOI: 10.3102/1076998609346961), and there are indeed methods to deeply assess the heterogeneity and publication bias (see Richardson et al. https://doi.org/10.1016/j.cegh.2018.05.005 and Ioannidis JPA, Trikalinos TA. The appropriateness of assymmetry tests for publication bias in meta-analyses: a large survey. CMAJ. 2007; 176 (8): 1091-1096. https:// 10.1503/cmaj.060410).

Based on my previous comments, there is a need to extensively revise the results, discussion, conclusion, abstract, keywords, article title and references. Just to add that in the discussion, it is important to start by presenting your main results that should be discussed with regard to data from the literature in the subsequent paragraphs, feasible recommendations to appropriate bodies should be made, and strengths as well as unavoidable limitations should be discussed.

2.2.2. Minor comments

As already said, extensive editing of the manuscript ideally with the aid of a native English speaker is warranted.

The abbreviation RCF is not conventional in Medicine, so it is not warranted.

Page 4 line 67, consider writing Data source instaed of search strategy

Page 4 line 68 Consider writing Medline (pubMed)

Page 4, lines 87-88: just say that you reported the risk estimates as mentioned in the primary studies articles

Page 4 line 82, please write "dat aextraction and mangement".

Revise the supplemental materials.

When revising, make sure there is no citation gaming (see Macdonald. https://doi.org/10.1177/05390184221142218) in the manuscript as this will be asessed.

Questions I find important for your SLR: can you compare the outcomes of residential care facilities with places for homeless elderlies (e.g. shelters...)? Does the status (migrant/elderly in his home country) affect the outcome ( I did not see interest for this in the manuscript)? I leave you with this reflection and this article: Om et al. BMC Geriatr

. 2022 Apr 25;22(1):363. doi: 10.1186/s12877-022-02978-9.

Finally, build a supplementary material with your work: see how others have done: Emmons-Bell et al. Heart

. 2022 Aug 11;108(17):1351-1360. doi: 10.1136/heartjnl-2021-320131.

Mickael Essouma

Available online on 28 March 2023

6. PLOS authors have the option to publish the peer review history of their article (what does this mean?). If published, this will include your full peer review and any attached files.

Reviewer #1: No

Reviewer #2: **Yes: **I have not signed this review on behalf of someone else.

---

## [Author Response · Author response to Decision Letter 0]

13 Apr 2023

Reviewers' comments:

Reviewer #1: 

The manuscript is based on a systematic literature review of an important topic, namely studies comparing either mortality or hospitalizations of older people living in residential care facilities versus nursing homes or communities. According to my evaluation the analysis has been carried out correctly and the manuscript is well written so I can recommend publication. The manuscript leaves the impression of only a limited number of available high quality studies and hence, the conclusion is wisely cautiously formulated. The Discussion is thorough and all central limitations are mentioned.

1. The only detail that caught my attention was that study 19 is classified as fair but nevertheless, is described as lacking data on age. In this research area I find that the age distribution is of central importance. 

Authors comment: We would like to thank the reviewer for her/his encouraging comments. Regarding study 19 (Mortality in older care home residents in England and Wales, Shah et al., PMID: 23305759), it is true that baseline characteristics of age were not described in this study. In fact, the study has compared “care home” versus “communities” with different age categories (65-74 years old, 75-84, 85-94 and 95-104). Nevertheless, further analysis have been provided with a distinction between “residential care” and “nursing care home” and the mortality has been compared between these two types of residences with adjustment on age and sex, and a further adjustment on dementia. It is why we have considered this study as “fair” and not “poor” according to our quality assessment methodology. We agree with you that age distribution is of central importance. 

2. Second, I mention probably a typo as abbreviations are consistently used but on page 10 for some reason 'nursing home' is written out, although the abbreviation NH has already earlier been introduced.

Authors comment: Apologize for this mistake. We have deleted the abbreviation NH throughout the manuscript. 

Reviewer #2: 

Major comment

Introduction

1. This is a global SLR, but not s SLR of COVID-19 or US-based data. So, there is no reason why you emphaisze on COVID-19 and USA in the introduction. I understand that the demand could have increased with the advent of the COVID-19 pandemic (West et al. Age and Ageing, Volume 50, Issue 2, March 2021, Pages 294–306, https://doi.org/10.1093/ageing/afaa289), but this could just be mentioned within the text as one among the many reasons why it is more and more important now to address the importance of residential care facilities when deciding about the place of care for elderlies.

Your introduction should not exceed one page, although it is important that you revise the current introduction providing information to these important questions (obviously for the global population), because PLOS One is a generalist not specialist journal: Is the health of older people a concern in the society? Notably, what is the proportion of older people in the current population and how will change in the coming decades? In general, how is the health state of older people: disease, disability-adjusted life years, years lived with disease, most frequent diseases, comorbidities, drug-consumption level? What are the available care delivery systems for older people? What is the place of residential care facilities in those delivery systems? How have the covid-19 pandemic and other stressors increased the demand for care delivery systems (including residential care facilities) in recent years? What should be done for a remedial to this alarming situation? How will your SLR help to remedy to the situation: is it intended to inform older people health experts? Stakeholders? End with a clear statement of the aim of your SLR. Writing and subsequent editing, will help you to deliver all those important messages within a short text of one page.

Authors comment: We thank the reviewer for this comment and we have now focused the introduction on the main recommended points within a short text of one page: 1. global data of older people and projections, 2. individual needs of older people, 3. available care delivery system, 4. lack of literature on residential care facilities and 5. clear statement of the aim on this systematic review. 

Methods

2. The reference 6 is irrelevant, because the registration number from PROSPERO is sufficiently informative.

Authors comment: Agree with reviewer’s comments. We have deleted reference 6.

3. Provide a reference for the PRISMA guidelines used. the latest one which is better to provide is: Page et al. BMJ 2021; 372 doi: https://doi.org/10.1136/bmj.n71.

Authors comment: We thank the reviewer’s for this recent reference that we have added this reference as requested.

4. The searched databases: I advise to include those that capture data for usually forgotten regions of Africa (e.g. AJOL, Africa Index Medicus), Asia (see DOI: 10.1097/EDE.0000000000000325) and South America (e.g. LILACS), as a global SLR should fairly provide data from across the globe. I would also include EMBASE (see Bramer et al. J Med Libr Assoc. 2018 Oct; 106(4): 531–541. doi: 10.5195/jmla.2018.283).

Authors comment: We agree with the reviewer’s comment that other databases coming from forgotten regions could have been studied. Nevertheless, nursing homes and residential care facilities are relatively underdeveloped in Africa and South America. An interesting analysis from the WHO (file:///C:/Users/423540/Downloads/9789241513388-eng.pdf) pointed out that published studies and reports of the long-term care comes from families in sub-Saharan Africa and are limited and skewed overwhelmingly to southern, western and eastern parts of the region, particularly to Ghana, Kenya, Nigeria, and South Africa. Organized systems of long-term care are generally lacking, families constitute the major source of care for older people who are no longer able to live independently. Published studies and reports of organized long-term care in sub-Saharan Africa are even more limited than the evidence base on family care. Most research comes from South Africa and concerns a particular sub-setting: residential facilities. Two major service models appear to dominate: charitable care for the most destitute older people (usually operated with few resources by faith-based, civil society or public welfare bodies) and private for-profit services, mostly in the form of residential homes for those who are able to pay. There appear to be few, if any, organized services for the majority of older people who fall between these extremes of the spectrum. 

Nevertheless, lack of nursing homes or residential care facilities in Africa is rather a good new to our point of view. These facilities are associated with unfavourable outcomes for older people. It is a mode of accommodation “by default” in the face of situations of major dependence and the organizations of the society. Moreover, nursing homes are very expansive (around 3000 euros per month). Is this model of accommodation for older people with dependency the one that should be replicated in Africa for example? We are not convinced, the covid-19 pandemic has shown the vulnerability of nursing home’s residents with a very high mortality. Other models exist and should be studied and developed, like foster families in the French Caribbean islands. We are working on this model and we consider that foster families could be an alternative to nursing homes (see comment 17). 

With the aging of the population, it will be a matter of interest in the future and we hope that we could do epidemiological studies in other countries than North American and Europe. Regarding Asia, we have included studies from Australia and South Korea. We propose to include this limit of our study on the discussion section.

5. The search terms should be revised. The search strategy should be based on three main concepts: the population of interest (elderly people: all the terms referring to this phrase and available in the searched databases should be sought), the exposure (residential care facilities again with all the terms referring to it, the nursing home with all synonyms, the home-based care with all synonyms) and the outcomes (hospitalizations with all synonyms such as admission, emergency visit; case fatality with all synonyms such as mortality and death). Did you perform hand searching? This should be clearly stated and explained in the manuscript. Work with an expert librarian.

Authors comment: we tried to do the most extensive research possible and reviewed over 7000 titles (and abstracts if necessary). We put a lot of thought into the best possible strategy, and we concluded that it is unlikely that we missed several studies if the terms “residential care” or “senior housing” or “independent living communities” or “continual care retirement” were not in the abstract, and the terms “mortality” or “hospitalizations” or “death” in the full text of the articles. We do not used the terms “nursing homes” or synonyms, nor the terms home base care or synonyms in our strategy. We could have narrowed our search with these keywords, with the risk of missing articles in which these words were not included. It is also true if we restrict our research to the population of interest as proposed. For example, if we add the filter “Aged: 65+ years” in our strategy on medline, we do not find the study of Inaccio et al. (Mortality in the first year of aged care services in Australia; PMID: 32815606) which is however relevant for our systematic review.

6. Inclusion and exclusion criteria: Do not place limits on the language of articles because this increases the selection bias, and the study is funded. So, you can request help from certified translators for articles written in languages which you are not familiar with, or use websites such as deepl translator. I am concerned by the fact that you excluded articles that focused on specific diseases, but you focused on dementia. Can you explain this inconsistency ? (which should be resolved throughout the manuscript). I am also concerned by the fact that you compare the outcome resulting from the care at residential care facility with that from original homes and nursing homes, but you exclude cross-sectional analytical studies that can also do this well: is there a justification? As stated above, it is also important that you clarify for authors the exposure and the outcomes assessed. Along this line, I would use the term "case fatality" throughout the text, instead of "mortality" (see Kelly and Cowling. Epidemiology 24(4):p 622-623, July 2013. | DOI: 10.1097/EDE.0b013e318296c2b6).

Authors comment: For the language, we agree with you and we have added this limit in the discussion section. We have excluded articles focused on specific mortality or hospitalizations, like suicide mortality, not articles from specific populations. We are interested by total mortality and total hospitalizations, not by specific mortality or hospitalizations that are not the topic of our study. We exclude cross-sectional studies since these studies cannot provide incidence data. We prefer the term mortality instead of case fatality in accordance with the scientific literature.

7. What do the IADL and ADL scores mean (page 4, line 86)?

Authors comment: The Instrumental Activities of Daily Living (IADL) scale (Lawton’s IADL scale) and the Activities of Daily Living (ADL) scale (Katz's scale) assessed functional status. We have added theses information and associated references as requested. 

8. The data extraction should be revised as well, and the standardized data abstraction sheet uploaded as supplemental material with your submission. These are mandatory data for high-quality epidemiological studies: type of sampling, setting [community versus hospital-based, registry], response rate [for surveys], locality [urban versus rural vs semi-urban], region of origin [based on which classification: world bank? UNSD?], country of origin, study design (please, go through this paper to gain an insight on types of traditional epidemiological observational studies . It should be noted that cohorts are always longitudinal studies, so the repetition in the text is not warranted), timing of data collection (retrospective/prospective/ambispective i.e., prospective + retrospective. Note that the mode of data collection is more important than when authors collected data i.e., in a registry with prospectively collected data but which can be retrospectively consulted by authors of a given manuscript, it is the fact that data were collected prospectively which is relevant; and this type of study is different from a retrospective chart review [Vassar and Holzmann. J Educ Eval Health Prof 2013, 10: 12 • http://dx.doi.org/10.3352/jeehp.2013.10.12.] where data were collected as usual in the clinic without aim to conduct a study, but subsequently some researchers conduct a study with those data that they therefore collect retrospectively).

Authors comment: The tables in the manuscript describe all the requested information: setting (nursing homes, residential care facilities or communities), country, study design, number of participants and outcomes. The locality ([urban versus rural vs semi-urban]) or region of origin are not useful or not available. Agree to delete the term “longitudinal”.

9. References 7 and 8 should be omitted because they are irrelevant and outdated. The tool used for quality assessment (choose anyone reliable that you are comfortable with among those provided here: Munn et al. Int J Health Policy Manag

. 2014 Aug 13;3(3):123-8. doi: 10.15171/ijhpm.2014.71.) should be clearly stated, and how you did the assessment should be illustrated in tables for each study, in the supplemental material.

Authors comment: We respectfully ask the reviewer to reconsider his position. The tool we used for this systematic review has been developed by the National Institution of Health (NIH) and this tool is far from irrelevant and outdated. It has been used in more than 300 reviews, including several recent studies like Argote M et al; Schizophrenia (Heidelb). 2022 Sep 29;8(1):78.PMID: 36175509 or Theoh et al., BMJ Open Diabetes Res Care. 2023 Feb;11(1):e003203, PMID: 36792169. The tool assessed the studies based on several criteria: (1) research objective; (2) study population and recruitment; (3) exposure measurement and assessment; and (4) statistical analyses methods. We have reflected beforehand on the major biases we have identified and believe that using another tool like ROBIN-I will lead to the same conclusions.

10. There is a need for a meta-analysis. The between-study heterogeneity is a conditional limitation of the meta-analysis (see Valentine et al. Journal of Educational and Behavioral Statistics April 2010, Vol. 35, No. 2, pp. 215–247 DOI: 10.3102/1076998609346961), and there are indeed methods to deeply assess the heterogeneity and publication bias (see Richardson et al. https://doi.org/10.1016/j.cegh.2018.05.005 and Ioannidis JPA, Trikalinos TA. The appropriateness of assymmetry tests for publication bias in meta-analyses: a large survey. CMAJ. 2007; 176 (8): 1091-1096. https:// 10.1503/cmaj.060410).

Authors comment: We have argued throughout the discussion that a meta-analysis is not clinically relevant due to 1. the methodological quality of the studies, 2. the lack of standard medical characteristics allowing us to adjust the analysis and 3. that the characteristics of residents are different between countries in terms of dementia. 4. Studies of good methodological quality describe opposite results to the others. Indeed, this systematic review pointed out the lack of good epidemiological studies allowing to compare nursing homes versus residential care facilities with similar clinical characteristics (co-morbidities and dementia).

11. Based on my previous comments, there is a need to extensively revise the results, discussion, conclusion, abstract, keywords, article title and references. Just to add that in the discussion, it is important to start by presenting your main results that should be discussed with regard to data from the literature in the subsequent paragraphs, feasible recommendations to appropriate bodies should be made, and strengths as well as unavoidable limitations should be discussed.

Authors comment: We have modified the manuscript as recommended.

Minor comments As already said, extensive editing of the manuscript ideally with the aid of a native English speaker is warranted.

12. The abbreviation RCF is not conventional in Medicine, so it is not warranted.

Authors comment: Agree, we have deleted the abbreviation RCF throughout the manuscript. 

13. Page 4 line 67, consider writing Data source instaed of search strategy

Authors comment: Agree.

14. Page 4 line 68 Consider writing Medline (pubMed)

Authors comment: Agree

15. Page 4, lines 87-88: just say that you reported the risk estimates as mentioned in the primary studies articles

Authors comment: Agree

16. Page 4 line 82, please write "dat aextraction and mangement".

Authors comment: Agree

17. Revise the supplemental materials.

When revising, make sure there is no citation gaming (see Macdonald. https://doi.org/10.1177/05390184221142218) in the manuscript as this will be asessed.

Authors comment: all authors have contributed to this work, we are the same team who work on residential care facilities and alternatives models to nursing homes. We just publish three articles on this matter:

1. Boucaud-Maitre D, Meillon C, Letenneur L, Villeneuve R, Dartigues JF, Amieva H, Tabué-Teguo M. Health trajectories of elderly living in senior housing: A Longitudinal Perspective. Sci Rep 13, 5471 (2023).

2. Boucaud-Maitre D, Cesari M, Tabue-Teguo M. Foster families to support older people with dependency: a neglected strategy. Lancet Healthy Longev. 2023 Jan;4(1):e10.

3. Boucaud-Maitre D, Villeneuve R, Simo-Tabué N, Dartigues JF, Amieva H, Tabué-Teguo M. The Health Care Trajectories of Older People in Foster Families: Protocol for an Observational Study. JMIR Res Protoc. 2023 Feb 8;12:e40604.

18. Questions I find important for your SLR: can you compare the outcomes of residential care facilities with places for homeless elderlies (e.g. shelters...)? Does the status (migrant/elderly in his home country) affect the outcome ( I did not see interest for this in the manuscript)? I leave you with this reflection and this article: Om et al. BMC Geriatr

. 2022 Apr 25;22(1):363. doi: 10.1186/s12877-022-02978-9.

Authors comment: Thanks for your proposal, we do not have this information in our studies but this could be the subject of another dedicated study.

19. Finally, build a supplementary material with your work: see how others have done: Emmons-Bell et al. Heart. 2022 Aug 11;108(17):1351-1360. doi: 10.1136/heartjnl-2021-320131.

Authors comment: the supplementary material contains the assessment of the quality of the studies of our systematic review. We thank the reviewer for the interesting reference and we have added a supplementary figure with the countries with Studies comparing residential care facilities versus nursing homes or communities on mortality and hospitalizations. As mentioned in the article, the studies comes from United States, Canada, England, Ireland, Australia and South Korea.

---

## [Decision Letter · Decision Letter 1]

12 May 2023

PONE-D-23-05379R1Comparison of mortality and hospitalizations of older adults living in residential care facilities versus nursing homes or communities. A systematic review.PLOS ONE

Dear Dr. Boucaud-Maitre,

Thank you for submitting your manuscript to PLOS ONE. After careful consideration, we feel that it has merit but does not fully meet PLOS ONE’s publication criteria as it currently stands. Therefore, we invite you to submit a revised version of the manuscript that addresses the points raised during the review process.

We look forward to receiving your revised manuscript.

Kind regards,

Charlotte Beaudart

Academic Editor

PLOS ONE

Additional Editor Comments:

As the editor assigned to your submission, I have carefully reviewed your responses to the reviewers' comments. While I believe your responses are appropriate, I would like to provide you with some additional feedback for further clarification. Please find my comments below:

There seems to be some confusion in the article regarding patients living in the community. The abstract and inclusion criteria are not clear in this regard. I only understood what you did in relation to this patient group when reading the later part of the results section.According to the PRISMA and AMSTAR2 criteria, authors are required to provide an example of their search strategy for at least one of the investigated bibliographic databases.Any limitations in the search process should be justified, including limitations on the language of articles. There are scientific references available that can be cited to support this argument (e.g., doi: 10.1017/S0266462312000086).Data extraction is typically expected to be performed in duplicate. Please acknowledge this as a limitation of the study.Has the assessment of study quality been conducted in duplicate?The "Statistical Analysis" paragraph appears to be incomplete. It is important to explain why a meta-analysis was not conducted.I would like the research question to be formulated using the PICO format.

Thank you for considering these comments and suggestions. I believe addressing these points will significantly improve the clarity and rigor of your article.

Reviewers' comments:

Reviewer's Responses to Questions

**Comments to the Author**

1. If the authors have adequately addressed your comments raised in a previous round of review and you feel that this manuscript is now acceptable for publication, you may indicate that here to bypass the “Comments to the Author” section, enter your conflict of interest statement in the “Confidential to Editor” section, and submit your "Accept" recommendation.

Reviewer #1: All comments have been addressed

2. Is the manuscript technically sound, and do the data support the conclusions?

Reviewer #1: Yes

3. Has the statistical analysis been performed appropriately and rigorously? 

Reviewer #1: Yes

4. Have the authors made all data underlying the findings in their manuscript fully available?

Reviewer #1: Yes

5. Is the manuscript presented in an intelligible fashion and written in standard English?

Reviewer #1: Yes

6. Review Comments to the Author

Reviewer #1: Nothing further to add. My few comments have been properly adressed. I have also read the second reviewer's comments and found that also they have been well adressed.

7. PLOS authors have the option to publish the peer review history of their article (what does this mean?). If published, this will include your full peer review and any attached files.

Reviewer #1: No

---

## [Author Response · Author response to Decision Letter 1]

16 May 2023

Comments to the editor:

As the editor assigned to your submission, I have carefully reviewed your responses to the reviewers' comments. While I believe your responses are appropriate, I would like to provide you with some additional feedback for further clarification. Please find my comments below:

1. There seems to be some confusion in the article regarding patients living in the community. The abstract and inclusion criteria are not clear in this regard. I only understood what you did in relation to this patient group when reading the later part of the results section.

Authors comment: We would like to thank the editor for raising these points in order to improve the quality of the article . We have compared the mortality and hospitalizations rates of older people living in residential care facilities or in the community. We agree with you and we have clarified this point in the abstract and inclusion criteria, using only the term “communities” throughout the article.

2. According to the PRISMA and AMSTAR2 criteria, authors are required to provide an example of their search strategy for at least one of the investigated bibliographic databases.

Authors comment: We have added an example of our strategy based on Pubmed in an additional file.

3. Any limitations in the search process should be justified, including limitations on the language of articles. There are scientific references available that can be cited to support this argument (e.g., doi: 10.1017/S0266462312000086).

Authors comment: We have added in the limitation section “It has been established that the exclusion of non-English language articles has only a minimal effect on the overall conclusions of the reviews” with the following recent reference:

Nussbaumer-Streit B, Klerings I, Dobrescu A, et al. Excluding non-English publications from evidence-syntheses did not change conclusions: a meta-epidemiological study. J Clin Epidemiol 2020;118:42–54.

4. Data extraction is typically expected to be performed in duplicate. Please acknowledge this as a limitation of the study.

Authors comment: We have added this limitation as suggested.

5. Has the assessment of study quality been conducted in duplicate?

Authors comment: the quality assessment of the study was conducted by a senior methodologist and not in duplicate. We have added this as a limitation.

6. The "Statistical Analysis" paragraph appears to be incomplete. It is important to explain why a meta-analysis was not conducted.

Authors comment: We agree with this comment and explain why we did not perform a meta-analysis in the statistical analysis: 

“After reviewing each of the studies included in this review, we considered that a meta-analysis was not relevant, due to the low methodological quality of the studies, the diversity of study designs and follow-up and the heterogeneity of populations, for which essential baseline characteristics were not available.”

7. I would like the research question to be formulated using the PICO format.

Authors comment: We agree with you and formulated the research question using the PICO format in the method section (eligibility criteria) as proposed by the PRISMA guidelines:

“Population: We only selected cohort studies conducted among older persons (≥65 years old). 

Intervention: We selected studies that had a cohort design (prospective or retrospective cohort design) and at least six months of follow-up and that measured mortality or hospitalizations as outcomes. For duplicate publications from the same cohort, we selected those with the largest number of participants. We excluded cross-sectional studies and, reviews. 

Comparison: We selected cohort studies comparing residential care facilities with nursing homes and/or communities,

Outcomes: To be included, we considered studies that reported the number of participants or person years and the number of deaths or hospitalizations in both groups (residential care facilities versus nursing home and/or communities). We excluded studies reporting mortality or a hospitalization for a specific medical condition.”

We hope that these changes meet your expectations.

---

## [Editor Report · Decision Letter 2]

18 May 2023

Comparison of mortality and hospitalizations of older adults living in residential care facilities versus nursing homes or the community. A systematic review.

PONE-D-23-05379R2

Dear Dr. Boucaud-Maitre,

We’re pleased to inform you that your manuscript has been judged scientifically suitable for publication and will be formally accepted for publication once it meets all outstanding technical requirements.

Kind regards,

Charlotte Beaudart

Academic Editor

PLOS ONE
---

## [Editor Report · Acceptance letter]

22 May 2023

PONE-D-23-05379R2 

Comparison of mortality and hospitalizations of older adults living in residential care facilities versus nursing homes or the community. A systematic review. 

Dear Dr. Boucaud-Maitre:

I'm pleased to inform you that your manuscript has been deemed suitable for publication in PLOS ONE. Congratulations! Your manuscript is now with our production department. 

Kind regards, 

on behalf of

Dr. Charlotte Beaudart 

Academic Editor

PLOS ONE